# Upper limb muscle reflexes in real and virtual environments: Insights into sensorimotor adaptations

Daniel D. Hodgson[1], Taha Butt[1], Brian H. Dalton[2], Tyler Cluff[1,3], Ryan M. Peters [1,3,4]*

**1** Faculty of Kinesiology, University of Calgary, Calgary, Alberta, Canada, **2** School of Health and Exercise Sciences, University of British Columbia Okanagan, Kelowna, British Columbia, Canada, **3** Hotchkiss Brain Institute, University of Calgary, Calgary, Alberta, Canada, **4** Department of Biomedical Engineering, University of Calgary, Calgary, Alberta, Canada

* ryan.peters1@ucalgary.ca

## Abstract

### Introduction

The mechanisms influencing neuromuscular adaptations in the upper limb within dynamic environments remain understudied, especially when exposed to altered visual and emotional conditions such as those simulated in virtual reality (VR). Here we utilize VR to manipulate visual feedback, inducing motion sickness and modulating sympathetic arousal, while assessing adaptations in sensorimotor integratin in the upper extremity using electrically evoked and muscle stretch reflexes.

### Methods

Eighteen healthy young adults experienced four experimental conditions while sustaining submaximal activation of their flexor carpi radialis (FCR) muscle by isometrically supporting a weighted load: baseline real-world (Pre-VR), stationary VR (VR-BL), dynamic VR with motion perception via a virtual rollercoaster ride (VR-C), and post-VR following re-entry to the real environment (Post-VR). Muscle activity was monitored via electromyography (EMG), while reflex activity was assessed using electrical (H-reflexes) and mechanically induced (noisy tendon vibration; NTV) reflexes in the FCR. Additionally, electrodermal activity (EDA) and psychosocial indicators of motion sickness (subjective questionnaires) were measured throughout.

### Results

H-reflex amplitude was suppressed during VR-C, which persisted into Post-VR; whereas NTV-reflexes were unaffected across conditions. Sympathetic arousal (e.g., EDA) and motion sickness symptoms increased significantly during VR-C compared to Pre-VR, but rapidly returned to baseline Post-VR. EMG within the target muscle (FCR) as well as in the brachioradialis was maintained across conditions, though

**Data availability statement:** All relevant data are within the manuscript and its Supporting Information files.

**Funding:** This research was funded by Natural Sciences and Engineering Research Council Discovery (RGPIN-2020-06068) and Idea-to-Innovation grants (I2IPJ 548811-2020), as well as a New Frontiers in Research Fund grant (NFRFR-2021-00199) awarded to RMP. DDH was additionally supported through a Natural Sciences and Engineering Research Council Doctoral Postgraduate Scholarship.

**Competing interests:** The authors have declared that no competing interests exist.

increased activation was observed in the biceps brachii beginning at the onset of VR immersion (VR-BL).

## Discussion

These findings suggest suppression of spinal excitability (H-reflex) when the perception of motion (VR-C) was added to a stationary VR experience. Meanwhile, muscle spindle sensitivity (NTV-reflex) remained consistent, highlighting potential fusimotor adaptations to maintain sensorimotor function under altered visual and emotional states. Persistent H-reflex suppression post-VR indicates lingering neuromuscular effects of immersive VR, underscoring the need for further exploration of VR's implications for rehabilitation and virtual training environments.

---

## Introduction

The ability to exhibit situation-appropriate motor control in dynamic environments is a testament to the intricate relationship between sensory inputs and motor outputs within the nervous system. This is orchestrated by a complex network of neural pathways, involving integration of sensory information (e.g., vision, proprioception, vestibular), under varying sympathetic and emotional states, to activate corresponding spinal and supraspinal neuromuscular outputs. While much research has focused on understanding these mechanisms within the lower limbs in the context of static balance [1,2], less attention has been given to the neural adaptations that occur in the upper limbs during dynamic experiences involving altered visual, sympathetic, and emotional states. This line of inquiry holds particular relevance for individuals who depend on refined upper limb control under stress, including athletes (e.g., climbers) and tactical professionals such as firefighters or military personnel.

Integral to motor control is sensory feedback from the periphery, such as from muscle and skin, which provides important information about the environment that can influence subsequent movement patterns. For example, myotatic stretch reflexes, mediated by muscle spindle receptors embedded within skeletal muscles, play a vital role in maintaining postural stability and coordinating muscle activity [3]. Muscle spindles are fundamental proprioceptors that encode changes in in muscle length and velocity via primary and secondary afferent signals [4–7]. Their unique efferent (fusimotor) innervation via gamma (γ) and beta motor neurons (MN) enables direct control over their sensitivity from supraspinal and spinal circuits. Fusimotor drive fine-tunes spindle sensitivity and modulates reflexes [3,8,9]. Given the confluence of both peripheral and central inputs arriving in the spinal cord, context-dependent adaptations of the resulting stretch reflex can arise from altered descending commands that modulate the excitability of spinal alpha motor neurons (α-MN), interneurons, and Renshaw cells, and/or from changes in muscle spindle sensitivity via γ-MN activation [3,10–13].

In experimental settings, stretch reflexes can be elicited using mechanical stimulation such as tendon taps or vibrations, including noisy tendon vibration (NTV)

[1,14,15]. The resulting muscle or tendon stretch from these techniques activates muscle spindles directly and assess the integrity of the full reflex arc, including spindle sensitivity and spinal excitability [16]. Alternatively, the Hoffmann (H) reflex is elicited via electrical stimulation of the peripheral nerve trunk, bypassing the muscle spindles and directly exciting the afferent neurons along their axon. As a result, H-reflexes primarily reflect spinal-level adaptations such as changes in α-MN excitability or presynaptic inhibition [1,17,18]. Since both reflexes share common spinal circuitry but differ at the initiation point, comparisons between NTV and H-reflex responses can be used to distinguish peripheral (e.g., fusimotor drive) versus central (e.g., spinal excitability) mechanisms of adaptation under varying experimental conditions [1,19,20].

In real-world environments, the excitability of the stretch reflex pathway appears to modulate alongside conditions such as postural configuration [18,21,22], the risk or consequence of a postural disturbance [23–25], and with sensory cues such as visual feedback [1,2]. For instance, soleus H-reflex amplitude was reduced when standing compared to sitting, even when the background muscle activity was equivalent between conditions [21]. In fact, H-reflex attenuation corresponds with altered postural stability such as when changing position from lying to sitting to standing [18,22] or from quiet standing to standing on unstable terrain [26]. Furthermore, Sibley et al. [23] demonstrated that H-reflexes decrease in amplitude when subjects stand at the edge of a height platform, suggesting potential alterations in pre-synaptic inhibition within the spinal cord. Other studies have shown similar findings of H-reflex suppression under conditions of heightened postural threat [24,25]. This may be an innate executive control existing to prevent excess movements or overreactions to trivial perturbations that could lead to harm.

Conversely, studies examining mechanically evoked short-latency stretch reflexes (SLR; 20–30 ms upper-limb latency) have reported facilitated responses during similar real-world threatening conditions [19,27,28]. For example, Horslen and colleagues [19] observed the strengthening of tendon stretch reflexes when either the consequence (subjects stood at the edge of a 3.2 m platform) or the likelihood (the platform was occasionally tilted) of a fall was increased. This suggests a favoring of postural reflexes when the cost or risk of falling is high. More recently, they used a similar experimental design to demonstrate the significant enhancement of (SLR) and medium-latency stretch reflexes (via ramp-and-hold stretches) and tendon-tap reflexes during a height-induced postural threat [28]. Furthermore, the SLR–velocity relationship increased by 36.1% during the high threat condition, providing evidence of heightened dynamic muscle spindle sensitivity during postural threat. While these studies were limited to reflexes in the lower limb during standing, similar findings have been reported in the upper extremities. For example, Nafati et al. [20] reported an attenuation of H-reflexes and an excitation of mechanical reflexes in the extensor carpi radialis during the performance of a cognitively demanding task, suggesting that spindle sensitivity may similarly adapt in response to increased cortical demand or postural threat. Enhancing muscle spindle sensitivity during periods of stress or uncertainty may represent some distinctive protective mechanism to maintain or elevate the overall integrity of the stretch reflex, improving the chance of a successful correction if necessary – for example, in response to increased task demands or heightened likelihood or cost of an error, such as the risk of injury from a fall [19].

Virtual reality (VR) presents a unique tool to study motor control in immersive and controlled environments. VR equips researchers to manipulate visual feedback and induce sympathetic responses (e.g., changes in electrodermal activity; EDA) in a safe and practical manner while monitoring neural responses in real-time. Altered postural control and physiological/psychological indicators of stress seem to manifest similarly in real and VR settings [29–32]. Cleworth et al. [29] used real and VR environments matched in visual scale and detail to compare changes in standing postural control, EDA, and psychosocial states (e.g., subjective fear and anxiety) while participants stood at real and virtual heights. They observed similar adaptations across all measures, highlighting the utility of VR in replicating real-world threats and similarly modulating physiological parameters.

VR environments that alter the visual experience and/or sympathetic arousal appear to influence spinal reflexes as well [1,2]. For example, Hodgson et al. [1] recently reported reduced H-reflex amplitude in virtual conditions of standing at both low and high heights compared to real environment standing at ground level, while NTV-induced reflexes

   

remained unchanged. We proposed that fusimotor drive may elevate muscle spindle sensitivity, thereby contributing to the maintenance of stretch reflex amplitudes when spinal excitability (H-reflex) is reduced. We also noted that altered visual environment alone was sufficient to suppress H-reflexes. In a similar study, Grosprêtre and colleagues [2] used VR to simulate standing at ground level, atop a building, and falling from the building. H-reflex amplitudes were significantly dampened during their most stressful condition, falling from the building, although in contrast to Hodgson et al. [1], they observed no difference in H-reflexes between standing in the real environment compared to standing in VR at either ground level or at height. Notably, this study did not monitor mechanically evoked reflexes. Despite this contradictory finding, complex visual states have been shown to depress H-reflex amplitudes [32–34] and increase muscle spindle activity [35,36].

VR has been reported to induce symptoms of motion sickness, sometimes referred to as VR sickness or cybersickness. Notably, VR sickness, as well as deficits in proprioception [limb position sense; 37] and cognitive speed [38], have been reported to persist immediately following return to the real environment. It is unclear whether spinal reflexes exhibit such persistence following the return from VR – applicable research seems to focus on adaptations when entering VR while overlooking the reemergence to the real environment [1,2].

Despite evidence that spinal reflex pathways adapt to changes in visual input and threat in the lower limb, relatively little is known about whether similar neuromuscular adaptations occur in the upper extremities under altered visual and sympathetic states. Understanding this is important for tasks requiring precise arm control under stress, such as climbing or tactical operations. Therefore, while progress has been made, the neuromuscular adaptations that occur in the upper extremities, especially in conditions altering visual feedback and sympathetic arousal (e.g., VR), remain incompletely described. The objective of this study was to characterize the neural adaptations of upper limb reflexes induced by manipulating the visual environment and sympathetic arousal. We used electrical (H-reflex) and mechanical (NTV) stimulation to discern global (spinal excitability) and fusimotor (muscle spindle-dependent) changes in reflex behaviour under these different visual feedback conditions. To appropriately address this objective, we aimed to use VR to investigate whether visual feedback, without (VR-BL) or with (VR-C) the perception of motion, alters physiological (electrodermal activity) or psychosocial (motion sickness questionnaire) indicators of stress compared to two conditions in the real environment (Pre-VR and Post-VR). It was predicted that: (i) similar to in the lower limb during conditions of elevated sympathetic stress [1], H-reflex amplitudes in the upper limb would attenuate in VR and become the weakest in the most stressful (VR-C) condition, (ii) mechanically evoked stretch reflex amplitudes would remain unaltered [1] under conditions of manipulated visual feedback and motion perception in the VR conditions, (iii) physiological indicators of stress and psychosocial indicators of motion sickness would be greatest in the VR-C condition, and (iv) VR-induced adaptations of physiological and psychosocial stress as well as the H-reflex would persist throughout the washout (Post-VR) condition upon returning to the real environment [37,38].

## Methods

### Subjects

18 healthy young adults (mean age: 19.2 years; SD: 1.4 years; 9 females) were recruited for this study using a convenience sampling method from the university population. Our sample size was selected based on previous sample sizes used for studies investigating H-reflexes during exposure to VR, which ranged from 15 [2] to 20 [1]. Participants entered the study with limited or no previous VR experience, were free from musculoskeletal injury in the upper limbs, and had no diagnosed neurological conditions which could impact experimental outcomes, such as multiple sclerosis or spinal cord injury. Written informed consent was obtained by all participants prior to taking part in this study. Recruitment for this study occurred from May to August 2024, and the University of Calgary's Conjoint Health Research Ethics Board approved all experimental protocols.

## Procedure

**Experimental setup.** Following informed consent procedures, participants were equipped with electromyography (EMG), EDA, and electrical stimulation electrodes, as well as a custom-built wearable tendon vibrator. Participants were then seated comfortably in a fully adjustable electrical reclining chair, manipulated such that their right arm rested on the armrest with a 90-degree angle at the elbow joint. The chair was adjusted forwards or backwards to achieve an appropriate distance from a hand crank lever, positioned next to the chair, which could be raised and held comfortably in their supinated right palm while maintaining a relaxed position (Fig 1A). Wrist angle and muscle activity in the FCR were monitored visually by the researchers throughout the experiment.

**Determination of background load level.** Consistent submaximal background FCR activation was encouraged to improve the signal-noise ratio of the H-reflex. Submaximal isometric contractions (10–30% of maximum activation)

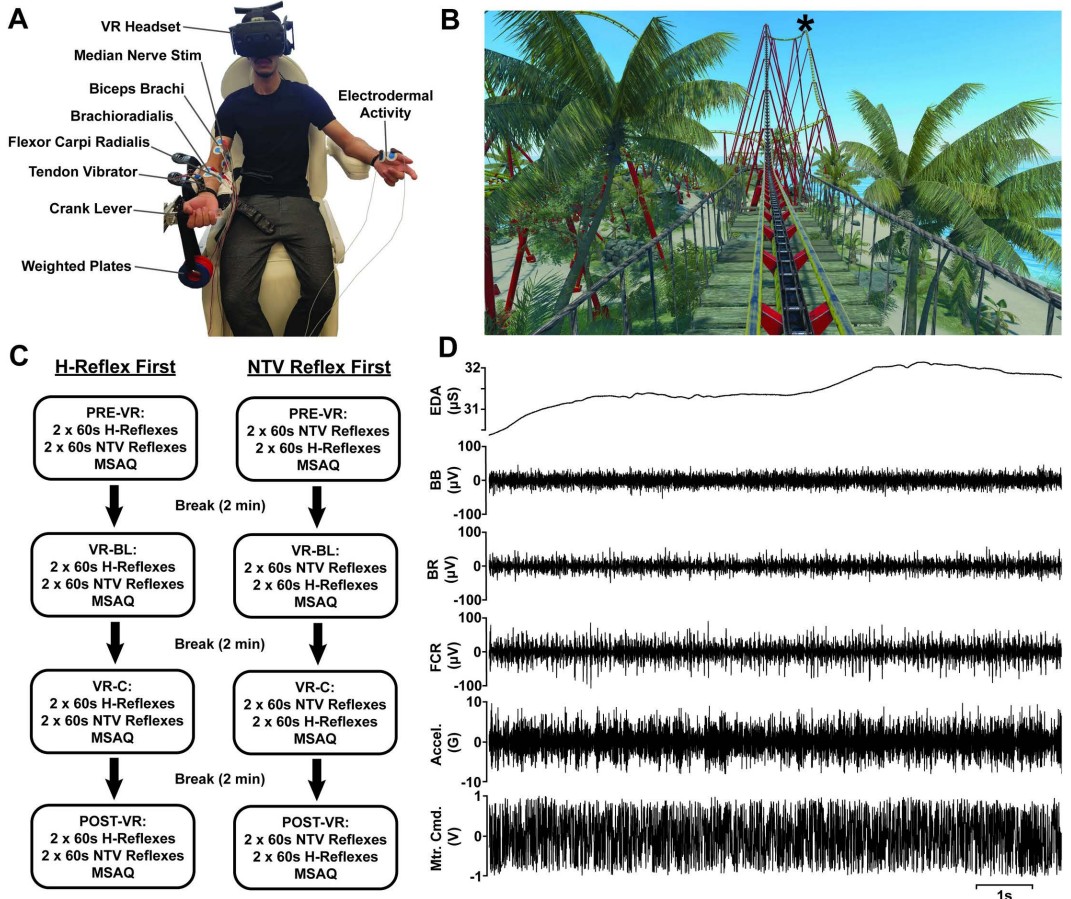

**Fig 1. Experimental setup and raw data.** A: Participants sat in an adjustable dental chair, donning a VR headset, EMG and EDA electrodes, nerve stimulation electrodes and a custom tendon vibrator. Background muscle activity was maintained by having them hold against a crank lever to counteract the torque applied from a set of pre-determined weighted plates. B: Schematic depicting the time course of the experimental conditions, breaks, and stimulus presentations with subjects for whom H-reflexes were presented first (left column) or second (right column). C: Example of the VR scene just prior to starting the rollercoaster, which was the scene for VR-BL condition. Asterisk at the crest of the first drop indicates the location where stimulation (H-reflexes or NTV) commenced. D: Exemplar 10s raw data traces during NTV stimulation, from top to bottom: Electrodermal Activity (EDA), Biceps Brachii EMG (BB), Brachioradialis EMG (BR), Flexor Carpi Radialis EMG (FCR), Tendon Vibrator Acceleration (Accel.) and motor command signal (Mtr. Cmd.).

have been shown to increase the successful observation of FCR H-reflexes, from 35% of participants at rest to 95% of participants during contraction [39]. It was crucial to obtain strong H-reflexes at the beginning of this study as their amplitude was predicted to decrease during the experiment. Improving the strength and success rate of observing an H-reflex enabled direct comparison of H-reflexes across conditions. This was achieved by placing an appropriate weighted load on the crank lever to be held by the participant. The load (eliciting 5–10% EMG%MAX in the FCR) was determined by first having participants perform three maximum voluntary contractions (MVC's), separated by 60 seconds, via manual muscle tests for the FCR: a researcher stabilized the participant's forearm in a fixed position against the armrest to isolate the wrist flexor muscles while applying a downward pressure on their supinated palm. Participants were instructed to exert maximal effort while flexing their wrist to counteract the downward pressure applied by the researcher. Each MVC effort was sustained for 3–5 seconds during which verbal encouragement was provided. EMG from the FCR muscle during the 3 MVC's was processed by removing the DC offset, rectifying, and taking the root mean squared (RMS) average (50 ms windows). The maximum EMG value was identified and represented 100% FCR muscle activation. Next, weighted plates (0.25–1 lb increments) were added to the crank lever until FCR EMG%MAX remained constant between 5–10% during a sustained isometric lever hold. This crank lever load was then used for the remainder of the experiment. Wrist angle was monitored for any deviations from this fixed position and verbal feedback was provided if adjustments were required.

**Determination of electrical stimulation current.** A one-time electrical stimulation recruitment curve procedure preceded the experimental procedure to determine the maximum peak-to-peak (P2P) amplitude and corresponding stimulus intensity values for the H-reflex ($H_{max}$) and M-wave ($M_{max}$) in the FCR muscle. Since these electrically evoked values vary between individuals, establishing an appropriate stimulation current intensity for the testing protocol specific to each participant is necessary. To accomplish this, participants were comfortably positioned in the adjusted chair with their arm relaxed at a 90-degree elbow angle. Their supinated right hand extended from the armrest and held the loaded crank lever (eliciting 5–10% FCR EMG%MAX) in a horizontal position while 1 ms square-wave pulses of electrical stimulation were delivered over the median nerve proximal to the elbow and medial to the biceps brachii muscle. Stimulations were delivered at least 10 seconds apart to mitigate risk of interference from presynaptic inhibition [18,40]. To limit fatigue, a researcher supported the crank lever, allowing the participant to relax between electrical pulse deliveries. Stimulation current began at 5 mA with subsequent 1–2 mA adjustments until an H-wave was detected for the FCR. Current increases continued until the H-wave amplitude began to descend with increasing current, at which point the current was dialed back by increments of 0.5–1 mA until the $H_{max}$ was determined. To locate the $M_{max}$, 2–5 mA current increments resumed until a plateau was observed in the rising M-wave. To ensure that maximal M-waves were consistently generated, 20% was added to the stimulation current required for producing $M_{max}$. This supramaximal stimulation current for the $M_{max}$ and the current required to elicit H-max were used for the electrical stimulation reflex assessments throughout the study. These recruitment curve procedures were completed within approximately 5 minutes, with 15–30 total electrical pulses delivered, and were followed by two minutes of rest prior to commencing the testing interventions.

**Testing interventions.** During the subsequent testing protocol, participants experienced four test conditions while maintaining a comfortable seated posture throughout. Each of the four conditions comprised four 60-second periods of stimulation: two periods of mechanical stimulation and two periods of electrical stimulation performed in randomized alternating order. Mechanical stimulation consisted of NTV over the FCR tendon near the wrist (evoking NTV-reflex) for the full 60-seconds; electrical stimulation was delivered to the median nerve using stimulation current intensities evoking H-reflexes and M-waves (see above). With approximately 10-seconds between deliveries, a total of seven electrical reflexes were elicited during each 60-second electrical stimulation period in randomized order: five targeting $H_{max}$ and two targeting $M_{max}$. The crank lever with a predetermined load (eliciting 5–10% of maximum FCR activation; see above) was held in a static horizontal position during all stimulation periods with 60-second breaks allotted between each epoch. Verbal feedback was provided by the researchers to correct for any deviations in crank lever position, which

was monitored visually. Each condition (including all four stimulation periods) was proceeded by the delivery of a Motion Sickness Assessment Questionnaire (MSAQ) [41] and the Borg CR-10 scale for measuring subjective rating of perceived exertion (RPE) [42].

**Experimental procedures and conditions.** The four conditions commenced with the first performed in the real environment ("Pre-VR"), followed by two conditions (one before and one during a virtual rollercoaster ride) in the VR environment (VR-baseline: "VR-BL", and VR-coaster: "VR-C"), and concluded with a final real-world condition ("Post-VR") (Fig 1B). A two-minute break separated each condition enabling the participant to release the crank lever and relax. Following Pre-VR, participants were introduced to the virtual environment. They donned an HTC Vive Pro 2 headset (HTC Corporation, Taiwan) with the fitment and view adjusted based on participant feedback until the clearest picture was achieved. A VR game "Epic Roller Coasters" was used for the VR environment, with the "Tropical Island" rollercoaster track selected for all participants. This track provides a rollercoaster ride approximately 60-seconds long, matching the timeframe for the stimulation periods in this study. Thus, participants completed four laps (e.g., 2× for H-reflexes and 2× for NTV reflexes) of the rollercoaster track during the VR-C condition. Upon transition into VR, participants were seated in a stationary rollercoaster cart at the beginning of the track, where an immersion protocol familiarized them with the virtual environment. They were asked to: 1) describe your surroundings, 2) locate the ocean to your right, 3) locate the large boulder to your left, 4) locate the treetops above you, and 5) locate the rollercoaster tracks beneath you. A monitor displaying the participant's field of view was positioned in the lab for reference. Following the familiarization protocol and a 60-second break, the two VR conditions commenced in sequence. VR-BL was performed first with the participant stationary at the beginning of the rollercoaster track (Fig 1C). The rollercoaster was then set in motion for the VR-C condition and the participant perceived their coaster cart climb an initial summit, taking approximately 60-seconds. The first, and each subsequent, stimulation period for the VR-C condition was initiated as the cart rounded the summit and began a steep rapid descent (see asterisk in Fig 1B). The remaining rollercoaster ride persisted approximately 60-seconds, incorporating fast-paced sharp turns, rises, drops, and loops, taking the rider around an interactive tropical island scene immersed with trees, birds, and ocean waves. Four laps were completed in sequence, with interleaved stimulation periods (NTV or electrical) each encapsulating one lap. Following the fourth and final lap, the participant returned to the starting position at the beginning of the rollercoaster track where the MSAQ was delivered, completing the VR-C condition. The VR headset was removed, reintroducing the participant back to the real-world environment to complete the final condition, Post-VR. Each of the four conditions lasted approximately eight minutes including four 60-second stimulation periods (two each of NTV and electrical), each separated by a 60-second rest period, and the MSAQ (~60-seconds).

## Data collection

**Electrical nerve stimulation.** Electrical stimulation of the median nerve was applied to generate H-reflexes and M-waves in the FCR muscle. The $M_{max}$ was used to monitor for changes in motoneuron activation (e.g., movement of a stimulating electrode) and peripheral excitability (e.g., sarcolemma excitability, neuromuscular propagation, or altered electrode position in relation to underlying muscle fibers) between conditions, which could confound H-reflex observations. Surface electrodes (MediTrace, Canada) were used for the anode and cathode. They were placed in sequence along the path of the median nerve, with the cathode approximately 5–7 cm proximal to the elbow joint just medial to the BB muscle belly and the anode centred 2 cm distally. A 3D printed rectangular block, custom designed to house the electrode connectors, was secured over the stimulation electrodes using a tensor bandage. By gently compressing the electrodes towards the median nerve, this method enabled optimal access and more consistent stimulation. A constant-current stimulator (STMISOLA, Biopac, USA) was used to deliver square-waveform electrical pulses (1 ms duration) using custom LabVIEW software (National Instruments, USA). Stimulator current (1–100 mA) was manually adjusted for each stimulus to the predetermined output for producing the desired physiological response.

 

**Noisy tendon vibration.** Muscle spindle-generated reflexes in the FCR muscle were evoked via mechanical stimulation of its tendon at the wrist. A custom-made wearable vibrator, secured to the distal forearm using an elastic strap (2 cm proximal to the wrist crease with the vibrating motor positioned along a direct line connecting the medial epicondyle and the scaphoid bone), was used to deliver NTV to the FCR tendon. NTV stimulation motor command signals were band-pass filtered white noise (4th order Butterworth filter; frequency cut-offs at 0.001 and 100 Hz), created with custom LabVIEW software (National Instruments, USA). The elastic strap was secured in position and held at a snug but comfortable tension prior to performing any study procedures and remained in place for the entire experiment. The device consists of a custom 3D printed housing with onboard motor (Haptuator BM3C, Tactile Labs, Canada), amplifier, and accelerometer (ADXL-354-CZ, Analog Devices Ltd., USA). Motor command signals were generated using custom LabVIEW software (National Instruments, USA) and output at 10 kHz from a real-time PXI system (PXIe-1062Q chassis; PXI-8105 embedded controller) with a multifunction data acquisition card (PXI-6289) and A/D board (BNC-2090).

**Electromyography.** Muscle activity was monitored throughout the study using surface EMG. Prior to surface electrode placement, the skin overlaying the muscles of interest was prepared as appropriate (e.g., shaved and cleaned with alcohol swabs). Two surface electrodes (Ambu, Malaysia) were applied to the skin, 2 cm centre-to-centre spacing, over the right FCR, brachioradialis (BR), and biceps brachii (BB). Electrodes were placed relative to the long axis over the thickest region of the muscle belly. For FCR, this was located approximately 1/3 of the distance along a direct line between the medial epicondyle and the radial styloid process. The BR electrodes were placed approximately 3 cm lateral to the FCR electrodes along the radial aspect of the antebrachium. The BB electrode position was approximately 1/3 of the distance from the cubital fossa to the acromion process in the midline of the muscle belly [43]. To ensure appropriate positioning of surface electrodes, muscle bellies were visualized and palpated during isometric contraction. A reference electrode was placed over the olecranon process. EMG was analog high-pass filtered at 30 Hz, amplified with a gain of 500 (NeuroLog NL844 pre-amplifier and NL820A isolated amplifier; Digitimer Ltd., England), and sampled at 5 kHz using a Power 1401 A/D board and Spike2 v10.13 (Cambridge Electronic Design, England). A 4th order low-pass IIR digital filter at 1000 Hz was applied offline to remove any high-frequency noise (MATLAB R2024a, Mathworks, USA).

**Electrodermal activity.** The galvanic skin conductance (e.g., sudomotor response) was monitored throughout the study by measuring electrodermal activity (EDA), providing a measurement of sympathetic arousal [44]. For each condition, EDA during NTV and electrical stimulation was assessed separately rather than pooled together to account for any influence of stimulation method on arousal. Two surface electrodes (Ambu, Malaysia) were placed on the thenar eminence of the left hand [1]. A Skin Conduction Unit (2502SA, Cambridge Electronic Design, England) recorded EDA with a sampling rate of 5 KHz using a Power 1401 A/D board and Spike2 v10.13 (Cambridge Electronic Design, England)

**Psychosocial assessments and questionnaires.** A Motion Sickness Assessment Questionnaire (MSAQ) was delivered following each condition [41]. The MSAQ consists of 16 statements pertaining to the subjective feelings of gastrointestinal (e.g., "I felt sick to my stomach"), central (e.g., "I felt faint-like"), peripheral (e.g., I felt sweaty"), and sopite-related (e.g., "I felt annoyed/irritated") distress experienced during the condition. Each statement was read aloud by a researcher to which the participant verbally expressed their agreement on a 1–9 scale (1 = "Not at all"; 9 = "Severely"). This questionnaire has been demonstrated to have high reliability under conditions inducive of motion sickness [41]. Scores from each of the four identified dimensions of motion sickness (gastrointestinal, central, peripheral, and sopite-related) were analyzed separately as well as concatenated to achieve an overall measurement of motion sickness. Additionally, the Borg CR-10 RPE scale [42] was employed to assess participants' perceived exertion during the study. This simple and widely recognized tool is commonly used to gauge exertion levels in exercise testing, training, and rehabilitation and has been validated against objective measures of exercise intensity [45].

## Data analysis

MATLAB (MathWorks, USA) and SPSS software (version 26.0; IBM SPSS Statistics Inc., USA) were used for all analysis with α set to 0.05. General linear mixed effects models (GLMM) were used to determine the main effect of visual condition on dependent variables. Fixed effects included Condition (Pre-VR, VR-BL, VR-C, Post-VR), while Participant was included as a random effect to account for repeated measures and inter-individual variability, thereby improving model precision and reducing the risk of inflated Type I error [46,47].

When a significant main effect of Condition was identified, we performed a focused set of pairwise comparisons using the Least Significant Difference (LSD) method, limited to planned contrasts based on a priori hypotheses [48]. This targeted approach strikes a balance between sensitivity to meaningful effects and appropriate statistical control. Planned comparisons included (i) Pre-VR vs. VR-BL, (ii) Pre-VR vs. VR-C, (iii) Pre-VR vs. Post-VR, (iv) VR-BL vs. VR-C, and (v) VR-C vs. Post-VR. These contrasts were selected to examine transitions into, within, and out of the VR environment, while comparing all conditions to the baseline (Pre-VR).

Dependent variables in both real and VR environments include mean EDA (assessed independently during NTV and electrical stimulation), mean MSAQ responses (collapsed into each of the four identified motion sickness dimensions as well as pooled altogether for a general indicator of motion sickness), rectified root mean squared (RMS) background EMG in each muscle (FCR, BR, and BB), H-Reflex/Mmax (H-reflex P2P amplitude expressed as a percentage of the Mmax P2P amplitude in the same condition) in the FCR, and NTV-reflex cumulant density P2P amplitude in the FCR. RMS is a commonly used measure of EMG magnitude and is consistent with reporting in our previous paper [1]. Cumulant density estimates are time-domain representations of the cross-spectral density computed between the input (vibration acceleration magnitude) and output (rectified EMG) signal, as described in detail previously [1,14]. Briefly, cumulant density, which is expressed as a unitless, time-dependent correlation, was estimated on a subject-by-subject basis for each visual condition by first concatenating BR EMG and acceleration signals across the two repeated 60-s intervals of NTV for each condition, and computing the inverse Fourier transform of the coherence between the input (acceleration magnitude) and output (BR EMG). We used a FFT segment length of 0.2048 s, which provided 585 segments and a resolution of 4.8828 Hz for each cumulant density estimate. The peak-to-peak amplitude was then extracted from the cumulant density and used in subsequent statistical analysis. For the input signal to all NTV analyses, we computed acceleration magnitude as the square root of the sum of squared x, y, and z axis accelerations (in units of G) [1]. Using vibration magnitude is favorable in this case, as the wearable device vibrates in three-dimensions and we sought to fully capture this motion – this is unlike previous studies that used bulkier motors which moved only in one axis [14].

Mean EDA during each type of stimulus delivery (electrical vs. NTV) was compared for each condition using paired t-tests to determine if participant arousal was consistent during H-reflex and NTV-reflex assessment. The mean time to peak for H-reflexes and NTV-reflex cumulant density will be measured as the latency between stimulus delivery and the maximum value of the initial peak of the bi-phasic reflex wave. Cohen's d effect sizes were calculated; magnitude of the effects may be interpreted as small (ES = 0.2), medium (ES = 0.5), or large (ES = 0.8) [49].

For post-hoc comparisons following a significant main effect in the GLMM, one-tailed tests were used for H-Reflex/Mmax (expected to decrease in VR conditions [Hodgson et al., Grosprêtre et al., Horslen et al.]) and EDA (expected to increase in VR conditions [1]). Two-tailed tests were used for all other dependent variables.

## Results

### General results

Two potential participants were screened from the study as we were unable to ascertain a consistent H-reflex in their FCR. This aligns with previous reports of a 72−100% success rate for evoking a response in this muscle [18]. Of the remaining 18 participants, all displayed reliable FCR H-reflexes via median nerve stimulation. Despite receiving primary

innervation from different nerves, occasional muscle responses to H-reflex were observed in the BR and BB muscles, although these were generally weak and inconsistent compared to those in FCR. Thus, we focused our H-reflex analysis on the FCR as planned. EDA data from one participant was excluded due to equipment malfunction during the experiment. Additionally, data points exceeding +/- 2SDs from the mean were considered outliers and removed. This resulted in the exclusion of a participant's data from four additional averages in the final analysis of EDA, Hmax, and BB EMG, one from each of the four experimental conditions. Each of these cases were irregularly high values. All remaining data points were within normal anticipated ranges consistent with previous research in our laboratory and were included in the analysis [1].

## Electrically evoked reflexes

**H-reflex/Mmax.** There was a significant main effect for condition when the H-reflex P2P amplitude was expressed as a percentage of the Mmax ($F_{(3, 64)}$ = 3.731; p = 0.016) (Fig 2). Significant pairwise contrasts indicated a sustained reduction in H-reflex strength beginning during the VR-C condition, which evoked reflexes smaller than both Pre-VR (25.2%; t(64) = 2.673; p = 0.005; CI: 2.642–11.421; d = 0.334) and VR-BL (20.0%; t(64) = 1.977; p = 0.026; CI: 0.908–9.589; d = 0.247) conditions. This reduction persisted upon return to the real environment, with Post-VR also registering smaller H-reflexes than Pre-VR (25.0%; t(64) = 2.651; p = 0.005; CI: 2.582–11.361; d = 0.331). There were no significant differences between real and VR baseline conditions (Pre-VR vs. VR-BL) or between VR-C and Post-VR. Pooled across conditions, the mean time to peak was 6.0 ms for the $M_{max}$ and 18.8 ms for the H-reflex.

**M-wave accompanying the H-reflex.** H-reflexes were typically accompanied by a small submaximal preceding M-wave which was monitored across conditions. The main effect for condition on the accompanying M-wave was not statistically significant ($F_{(3, 63)}$ = 2.143; p = 0.104).

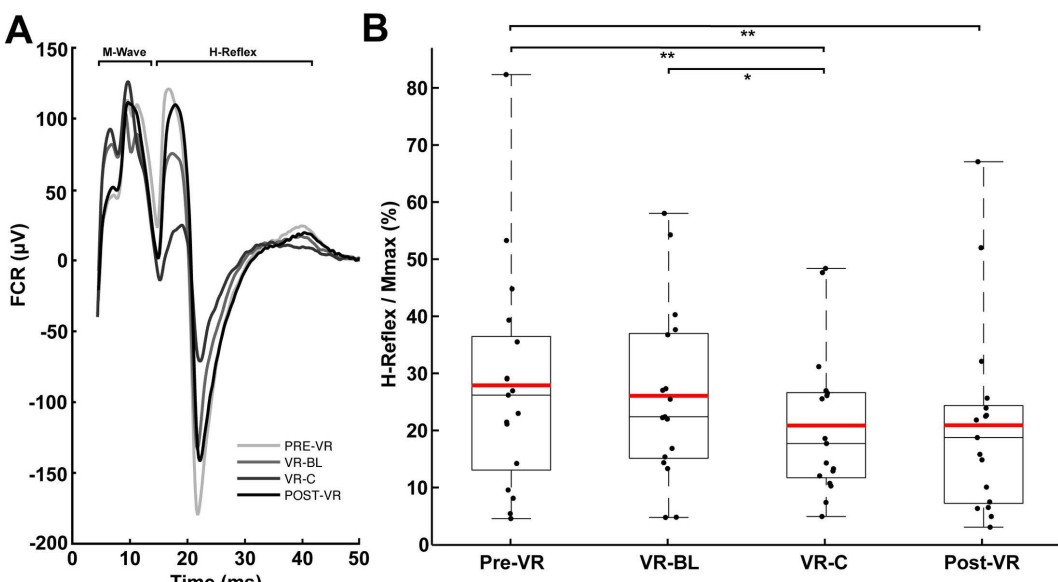

**Fig 2. H-reflex results across visual conditions.** A: Example of averaged H-reflex responses from a representative subject across all visual conditions, showing the trend for reduced H-reflex amplitude in the VR-C condition, the artifact at time zero, and the H-reflex occurring at a latency of ~20 ms. B: Mean H-reflex peak-to-peak amplitudes for each subject across visual conditions. Boxplots show the interquartile range (IQR), red lines indicate mean values, and error bars represent the full range of observed values for each condition. For all figures, asterisks denote significance at the p = 0.01 (*), 0.01 (**), and 0.001 (***) levels.

## Noisy tendon vibration reflexes

There was no significant main effect for condition on the P2P amplitude of the NTV-reflexes (F(3, 68) = 0.78; p = 0.509) (Fig 3). The mean time to peak after stimulus of NTV-reflexes was 24.6 ms and no differences were observed between any experimental conditions.

## Background muscle activity

The FCR was loaded with a weight targeting 5–10% of its EMG%MVC to optimize the signal-noise ratio of the reflex responses and to avoid the confounding factor of changes in muscle activity across conditions. Mean background EMG%MVC in the FCR was 8.1% and there were no differences between conditions (F(3, 68) = 0.732; p = 0.536) (Fig 4). Thus, the target muscle remained consistently active throughout the experiment. Similarly, there were no differences in EMG across any conditions for the BR conditions (F(3, 64) = 1.092; p = 0.359). However, there was a main effect for condition on BB EMG (F(3, 64) = 4.018; p = 0.011). Pairwise contrasts demonstrated significantly greater BB EMG in all conditions compared to Pre-VR, including VR-BL (42.3%; t(64) = 2.712; p = 0.009; CI: 4.11–27.11; d = 0.67), VR-C (47.3%; t(64) = 3.036; p = 0.003; CI: 5.98–28.97; d = 0.75), and Post-VR (42.2%; t(64) = 2.705; p = 0.009; CI: 4.07–27.07; d = 0.67).

## Electrodermal activity

Mean EDA for each condition was ascertained during the assessment of NTV and H-reflexes independently to account for any influence of stimulation method on arousal. While mean EDA, averaged across each 60-second reflex-stimulation period, exhibited a similar trend across conditions, it was higher during electrical (H-reflex; M = 14.5 µS, SD = 7.5 µS) compared to mechanical (NTV; M = 13.4 µS, SD = 7.43 µS) stimulation (8.2%; t(16) = 2.5, p = 0.025, d = 0.635) when pooled across all conditions. The electrical stimulation itself may have transiently elevated EDA in some participants creating this discrepancy. To minimize potential confounding effects caused by the H-reflex test stimulus, we focus on reporting EDA measured during NTV delivery which was more consistent throughout trials and across participants. As demonstrated in

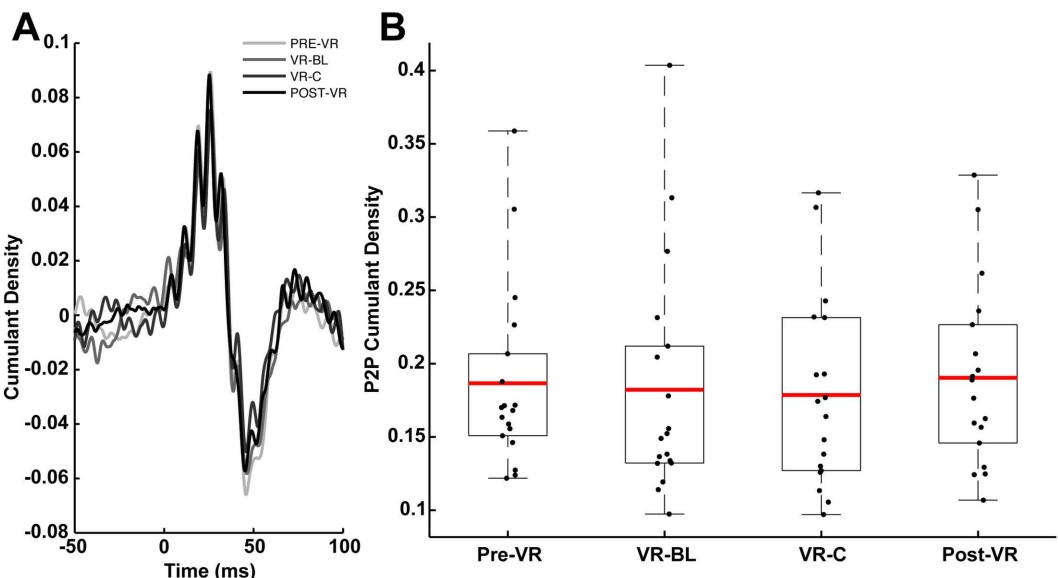

**Fig 3. Noisy Tendon Vibration (NTV-reflex) results across visual conditions. A:** Example of NTV responses from a representative subject across all visual conditions resulting from 120 s of vibration, showing a biphasic, short-latency response occurring within latencies of ~20-50 ms. **B:** Mean NTV peak-to-peak cumulant density for each subject across visual conditions. All other notations are identical to Fig 2.

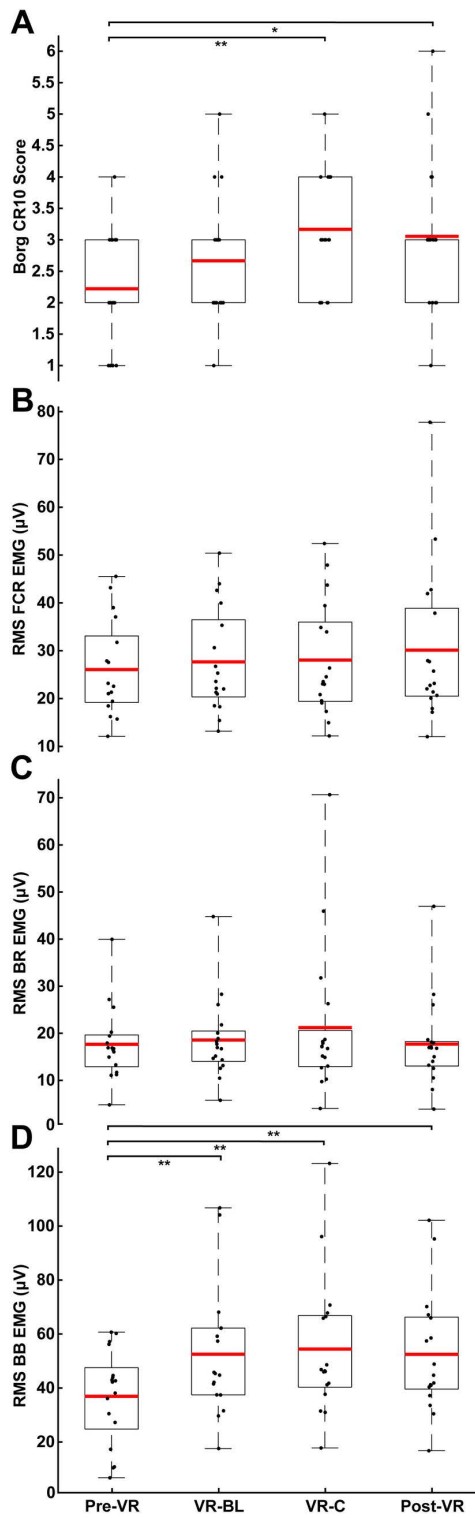

**Fig 4. Perceived fatigue level and background muscle activity across visual conditions. A:** Borg scale CR10 scores for each condition. **B-D:** RMS background muscle activity across each condition for FCR ('target muscle'), BR, and BB, respectively. All other notations are identical to Fig 2.

(Fig 5), EDA was elevated in all conditions compared to the real environment baseline condition (Pre-VR), reaching significance when the virtual rollercoaster was set in motion (VR-C; 42.6%; t(60) = 2.543; p = 0.007; CI: 1.608–7.77; *d* = 0.63) and persisting after returning to the real environment (Post-VR; 30.2%; t(60) = 1.780; p = 0.040; CI: 0.205–6.443; *d* = 0.45).

## Psychosocial assessments and questionnaires

The MSAQ questionnaire was administered at the conclusion of all four conditions. There was a significant main effect for condition (F(3, 68) = 11.9; p < 0.001) for overall motion sickness (all 16 items combined). When divided into separate questionnaire components, significant main condition effects were discovered for each of gastrointestinal, central, peripheral and sopite-related indicators of motion sickness. For the pooled result, as well as for each separate component, pairwise contrasts indicated significantly elevated motion sickness when both visual and motion perception were modulated during VR-C compared to all other conditions. Notably, there was no difference in overall, nor in any particular indicator of motion sickness, between the two real (Pre-VR and Post-VR) conditions, indicating that these symptoms did not persist upon re-emergence from VR. MSAQ responses are detected in Fig 5 and statistical differences are summarized in Table 1.

Finally, there was a significant main effect on condition for subjective muscle exertion reported using the Borg-scale (F(3, 68) = 3.368; p = 0.023). Pairwise contrasts revealed that significant elevations in perceived muscle exertion occurred during both VR-C (t(68) = 2.864; p = 0.006; CI: 0.286–1.602; *d* = 0.70) and Post-VR (t(68) = 2.527; p = 0.014; CI: 0.175–1.491; *d* = 0.62) compared to Pre-VR.

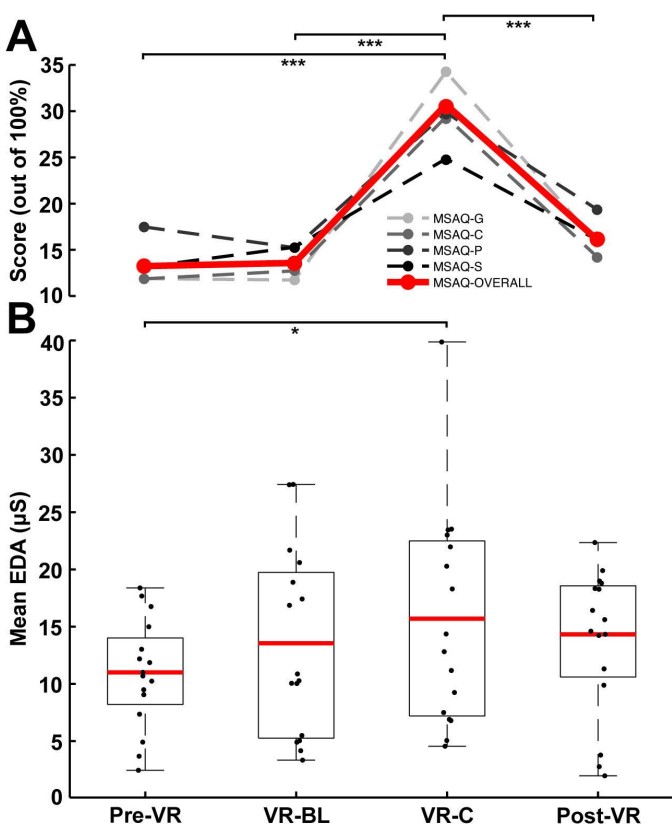

**Fig 5. Perceived motion sickness and sympathetic activity across visual conditions. A:** Mean MSAQ questionnaire scores for each subsection (grey scale) and combined (red). **B:** Mean EDA level across each of the visual conditions. All other notations are identical to Fig 2.

**Table 1. Summary of significant pairwise comparisons for perceived motion sickness (Motion Sickness Assessment Questionnaire; MSAQ) and muscle exertion (Borg CR-10 scale). MSAQ data are pooled (Overall) and separated into four indicators of motion sickness: gastrointestinal (G), central (C), peripheral (P), and sopite-related (S).**

| Measure | Main Effect | Condition Comparison | % Change | t(68) | p-value | Confidence Interval (CI) | Effect Size (Cohen's d) |
|---|---|---|---|---|---|---|---|
| **MSAQ – Overall** | (F(3, 68) = 11.9; p < 0.001) | VR-C – Pre-VR | 132.1% | 5.132 | < 0.001 | 10.68–24.27 | 1.24 |
| | | VR-C – VR-BL | 126.1% | 5.03 | < 0.001 | 10.33–23.93 | 1.21 |
| | | VR-C – Post-VR | 91.3% | 4.305 | < 0.001 | 7.87–21.46 | 1.10 |
| **Gastrointestinal Indicators (G)** | (F(3, 68) = 11.096; p < 0.001) | VR-C – Pre-VR | 188.3% | 4.911 | < 0.001 | 13.28–31.47 | 1.20 |
| | | VR-C – VR-BL | 192.1% | 4.945 | < 0.001 | 13.44–31.62 | 1.21 |
| | | VR-C – Post-VR | 117.7% | 4.064 | < 0.001 | 9.45–27.61 | 1.05 |
| **Central Indicators (C)** | (F(3, 68) = 11.304; p < 0.001) | VR-C – Pre-VR | 147.9% | 5.029 | < 0.001 | 10.58–24.49 | 1.47 |
| | | VR-C – VR-BL | 131.1% | 4.781 | < 0.001 | 9.71–23.62 | 1.39 |
| | | VR-C – Post-VR | 107% | 4.356 | < 0.001 | 8.23–22.14 | 1.34 |
| **Peripheral Indicators (P)** | (F(3, 68) = 5.5; p = 0.002) | VR-C – Pre-VR | 70.6% | 3.167 | = 0.002 | 4.57–20.13 | 0.83 |
| | | VR-C – VR-BL | 96% | 3.748 | < 0.001 | 6.83–22.39 | 1.00 |
| | | VR-C – Post-VR | 54.3% | 2.692 | < 0.001 | 2.72–18.27 | 0.75 |
| **Sopite-related Indicators (S)** | (F(3, 68) = 7.028; p < 0.001) | VR-C – Pre-VR | 89.4% | 4.255 | < 0.001 | 6.23–17.23 | 1.18 |
| | | VR-C – VR-BL | 62.6% | 3.472 | = 0.001 | 4.07–15.07 | 0.94 |
| | | VR-C – Post-VR | 54.8% | 3.192 | = 0.002 | 3.30–14.30 | 0.95 |
| **BORG CR-10** | (F(3, 68) = 3.368; p = 0.023) | Pre-VR – VR-C | | 2.864 | = 0.006 | 0.29–1.60 | 0.70 |
| | | Pre-VR – Post-VR | | 2.527 | = 0.014 | 0.18–1.49 | 0.62 |

## Discussion

### Summary

The primary purpose of this study was to use VR to manipulate visual feedback and sympathetic responses and examine how H-reflexes and NTV-induced reflexes adapt in the upper limb under varying sensory conditions. We observed a substantial weakening of the H-reflex during the dynamic VR condition (VR-C), coinciding with increased motion perception and physiological arousal. This suppression persisted after returning to the real environment, suggesting a lingering neural adaptation. Meanwhile, NTV-reflexes remained steady throughout all manipulations. Background EMG in the FCR and BR was unchanged across conditions, although the BB became more active following VR immersion, reflecting increased stabilizer engagement. EDA and motion sickness ratings confirmed elevated sympathetic and perceptual stress during VR-C, with EDA increases persisting Post-VR. Together, these results show that dynamic visual cues and sympathetic arousal modulate spinal reflex excitability in the upper limb.

### H-reflex and NTV-reflex modulation

The primary objective of this research was to examine the neuromuscular adaptations to the altered sensory feedback presented with both stationary and motion-simulating VR experiences. Interestingly, H-reflex strength was not impacted by transitioning into a stationary VR scene (VR-BL) but was significantly decreased when the virtual rollercoaster was set in motion (VR-C), an adaptation that persisted beyond the transition back into the real environment (Post-VR). Meanwhile, NTV-reflexes remained constant during transition into VR, throughout the VR experience, and after re-emergence into the real environment.

**H-reflexes.** In contrast to our prediction, the H-reflex was consistent between Pre-VR and VR-BL conditions. This is contrary to our previous finding, which highlighted that transitioning into a stationary VR environment while standing

was sufficient to supress soleus H-reflex strength [1]. It was postulated that, given the otherwise similar nature of the two conditions, the neural control of standing is altered in VR. Considering the present findings though, it seems that this adaptation may be task or posture dependent –the loaded upper limb muscles here, while active, did not contribute to maintaining balance in a seated position as the lower extremity muscles did during standing. So, the neural adaptations that occur alongside the change in visual environment with VR immersion may be task or muscle dependent, or at least primarily targeted toward the body regions most involved in the task. However, one other study has reported on H-reflex adaptations using VR and they observed no differences between real and virtual conditions, despite measuring from the soleus during stance [2]. Overall, while complex visual states can suppress H-reflex amplitudes in the lower limb during standing [32–34], our data show this does not generalize automatically to upper limb reflexes during seated tasks.

Consistent with our prediction, H-reflex amplitude was significantly reduced during the VR-C condition compared to Pre-VR. This finding aligns with previous studies which have demonstrated H-reflex suppression during their most arousing conditions, including lower limb tasks in both real [24,25] and virtual [1,2,29,30,32] environments, as well as upper limb tasks during cognitively demanding conditions [20]. This general agreement indicates that enhanced visual stimulation, motion perception, and sympathetic arousal can downregulate reflexive motor responses, likely via stronger descending spinal inhibition. In the current study, this adaptation occurred in the upper limb and coincided with the introduction of motion perception to the stationary VR scene and concurrent elevations in subjective motion sickness and sympathetic arousal, reinforcing the link to perceived threat. Thus, the reduction in H-reflex amplitude during VR-C may reflect a protective mechanism to limit unnecessary or exaggerated movements when a threat (e.g., a moving rollercoaster, fear of falling, or motion sickness discomfort) is perceived. This modulation could affect fine motor control in situations requiring precise upper limb stability, such as in athletes, surgeons, or operators using VR for training.

Importantly, H-reflex suppression persisted even after returning to the real environment (Post-VR). To our knowledge, this is a novel finding, as most studies focus only on reflex changes during VR exposure itself. This sustained effect suggests a residual effect of the VR experience on the CNS. Although fatigue could play a role, the incomplete recovery of EDA indicates that elevated sympathetic arousal may help maintain this suppression after VR ends.

**NTV-reflexes.** Aligning with our prediction, NTV reflexes remained constant during the transition from the real environment to both stationary and dynamic VR conditions, as well as post-VR. This aligns with our previous finding of stable NTV-reflexes in the soleus during standing across real and virtual conditions despite varying visual feedback and perceived threat [1]. In that work, stable NTV-reflexes coincided with weakened H-reflexes, suggesting that muscle spindle sensitivity becomes enhanced to preserve overall stretch reflex integrity when spinal excitability is downregulated. A similar trend has emerged in the current study, particularly during VR-C and Post-VR, when H-reflexes were weakened while NTV-reflexes remained unaffected. This supports the idea that fusimotor drive helps maintain reflex function under elevated threat or arousal.

## Background muscle activity and neuromuscular fatigue

To standardize background EMG in the FCR, participants supported a crank lever loaded to elicit 5–10% of their maximal FCR activity. We observed a mean of 8.1% maximal FCR output with no significant differences between conditions, indicating stable muscle activity throughout the experiment. Consequently, no differences in EMG were noted in the FCR or BR, indicating that the observed changes in the H-reflex were not heavily influenced by the level of muscle activity in these muscles.

However, the BB showed a significant and sustained increase in EMG after baseline (Pre-VR), persisting through all VR conditions and Post-VR. This may be attributed to increased tensing or stabilizer engagement due to sensory perturbation of VR. Although co-contraction requires activation of an agonist-antagonist pair and was not directly measured, increased stiffness has been observed when postural control is either compromised [50] or threatened [51,52], and with visual manipulations such as emotion-provoking images [53,54] and VR [1]. However, most of these studies report

increased co-contraction of muscles primarily involved in the task (e.g., lower limb muscles during stance). Here, the primary muscle for the task of wrist flexion (FCR) was not impacted, rather, the increased contraction occurred in a stabilizer muscle (BB). Notably, stabilizing muscles such as the BB can play a role in anticipatory postural adjustments, which occur before voluntary movement and help maintain stability under conditions of mechanical or environmental change [55]. The persistent BB activation may therefore represent preparatory stabilization in response to VR-induced sensory changes. Since we did not measure muscle stiffness, co-contraction, or H-reflexes in the BB directly, this interpretation remains speculative.

An alternative explanation is that BB activity compensated for mild fatigue in the FCR to maintain task performance. Submaximal tasks can induce fatigue without a reduction in task performance, as additional motor units or muscles are recruited [56]. Here, while additional motor units did not appear to have been recruited by the FCR (as would be indicated by greater EMG), BB EMG increased along with perceived exertion during VR-C and Post-VR. This suggests a strategy of greater stabilizer contributions to overcome sustained submaximal workloads as the experiment progressed. Notably, BB does not contribute to wrist flexion but does help maintain supination, which was required throughout the task. It is possible that BB became more active as the supinator muscle grew fatigued. Furthermore, participants were focused on maintaining FCR activation steady, which may have encouraged selective engagement of stabilizers like BB to maintain posture. Since BB activity increased alongside the transition into VR and prior to any significant change in perceived exertion, it likely reflects both a tensing response to the sensory manipulation and a compensatory adjustment as the task continued, while FCR and BR activity remained stable.

### Physiological and psychosocial modulation

Physiological and psychosocial arousal were manipulated by altering visual feedback and the perception of motion within VR. Significant increases in EDA emerged during the VR-C condition suggesting increased sympathetic arousal. This aligns with prior findings that VR-induced threat can induce sympathetic responses including elevations in skin conductance, heart rate, and subjective fear and anxiety perception similar to real environment threatening scenarios [29,31]. Notably, there was no appreciable change in EDA when transitioning between visual environments (real to VR and vice versa). The absence of a significant change upon initial immersion into VR during the VR-BL condition aligns with previous studies reporting either minor or no significant changes in sympathetic arousal when entering virtual environments [1,29,57]. After exiting the virtual domain, sympathetic activity in the real environment remained significantly elevated. EDA recovery following a visual manipulation such as VR exposure has not been previously reported on. The invariance of EDA during transitions into and out of VR suggest that altered physiological stress did not factor into any potential neuromuscular adaptations alongside these transitions, instead implicating the influence of the visual manipulation. However, since EDA was significantly elevated alongside the addition of virtual motion during VR-C, sympathetic arousal and/or emotional state may contribute to any accompanying alterations in reflex behaviour *within* VR.

Following a similar pattern, self-reported symptoms of motion sickness were vastly elevated during VR-C. This was true both for overall MSAQ scores as well as each of the four independent domains of motion sickness. Dissevering this result further reveals three noteworthy findings: (i) similar to EDA, motion sickness was unaffected by the transition into static VR; (ii) also aligning with EDA, motion sickness was strongly elevated when motion was introduced to the VR scene; and (iii) in contrast to EDA, motion sickness recovered quickly after returning to the real environment. This partially aligns with previous studies showing VR can induce VR sickness, although in this study it was the addition of motion, not VR in isolation, that triggered symptoms. Furthermore, in contrast to previous work [37], VR sickness evoked in the current study resolved rather quickly following re-entry into the real environment.

Recent studies have discussed the possibility of a significant correlation between motion sickness and EDA [58,59]. This association appears to be appropriate, although the timing of data extraction is an important consideration. Tracking VR sickness using EDA seems most suitable when users are most immersed in the situation rather than at rest or near

the starting or end point of the VR exercise [58]. This helps explain the coupled responses of EDA and VR sickness during VR-C, but their divergence during recovery in Post-VR. These physiological responses highlight the importance of considering residual arousal when designing VR-based rehabilitation or training programs, to minimize unintended effects on motor performance.

**Neural mechanisms for reflex modulation**

Both vibration-induced reflexes (mechanical) and H-reflexes (electrical) are understood to involve short-latency responses that rely on similar spinal circuits, though they differ primarily in their activation mechanisms and location. Vibration-induced reflexes are triggered by mechanical stretch of the muscle or tendon, engaging the entirety of the Ia afferent pathway via muscle spindle activation. The H-reflex, in contrast, is elicited through direct stimulation of Ia afferent fibers along their journey toward the dorsal horn, therefore bypassing the initial component of the circuit, the muscle spindle. Thus, the H-reflex pathway is incomplete compared to mechanically induced reflexes. Given this distinction, contrasting patterns in responses of these reflexes have been attributed adaptations specific to the muscle spindle such as altered spindle sensitivity, presumably driven by changes in fusimotor activity.

Despite some caution to this approach [60], numerous researchers have inferred fusimotor activity by examining how mechanical and electrical reflexes adapt across conditions. Burke et al. [60], caution against comparing them *within* a condition due to potential confounding factors; however, observing their adaptive patterns *across* conditions enables inferences as to the neural origin of these adaptations to be made with greater confidence if experiments are well controlled. In our study, the contrasting behavior of H- and NTV-reflexes during VR-C and Post-VR supports the idea that fusimotor drive maintains spindle sensitivity to offset reduced spinal excitability under elevated threat or arousal [1].

Prior research shows similar patterns under postural threat. For example, Sibley et al. [23] noted reduced H-reflex amplitudes when subjects stood on the edge of a height platform, postulating increased presynaptic inhibition or fusimotor drive. This is supported by subsequent research citing altered spindle sensitivity in conditions of height-induced postural threat via increased stretch [19,27,28], or decreased [1,2,24] or unchanged H-reflexes [19]. In our recent study [1] we found that H-reflexes were smaller in virtual environments simulating standing at varying heights, while vibration-induced reflexes remained stable. We suggested that fusimotor drive may help to maintain spindle sensitivity even when spinal excitability, as measured by the H-reflex, is diminished. Interestingly, in the current study, H-reflex strength was not impacted by transitioning into stationary VR (VR-BL) but was significantly decreased when the virtual rollercoaster was set in motion (VR-C), an adaptation that persisted beyond the transition back into the real environment (Post-VR). Meanwhile, NTV-reflexes remained constant during transition into VR, throughout the VR experience, and after returning from VR. So, the two reflexes manifest very differently during identical conditions prior to and following VR exposure. The contrasting H- and NTV-reflex behaviours arising during VR-C and ensuing through Post-VR supports that neural adaptations specific to muscle spindle circuitry may exist to enable peripheral feedback to compensate for the dampening of overall spinal excitability that seems to arise in conditions of altered visual and/or sympathetic state [1]. This also demonstrates that VR and/or threat induced spinal inhibition may persist for an undetermined period following removal of the stimulus.

The relationship between fusimotor and skeletomotor activity is often discussed as either *α-γ* co-activation or as partial fusimotor independence [10–13,61,62]. In the current study, FCR EMG did not change during the VR-C and Post-VR conditions despite the presumed increase in spindle sensitivity, implying *α-γ* independence. Human microneurography studies support this: direct recordings show fusimotor modulation, without corresponding changes in extrafusal activation, during passive movement tasks requiring focused attention [63,64], active movements during visuomotor learning [35,36], and in stationary muscles while preparing for movement [65].

Centrally, dynamic and static *γ*-MN are regulated by brain regions including the mesencephalon, basal ganglia, cerebellum, and motor cortex [3]. Peripherally, fusimotor control is shaped by sensory feedback from afferent reflexes originating in muscles, skin, and joints. Descending control enables the nervous system to adapt spindle sensitivity in response to varying

motor tasks, sensory inputs, and psychological states, based on volition. Meanwhile, peripheral influences may enable rapid responses to situational factors encountered within the environment. Together, such dynamic modulation likely plays a crucial role in fine-tuning both reflexive and volitional motor actions, ensuring precise motor control during tasks involving postural threat, altered visual feedback, or heightened arousal, independent from EF fibre activity and spinal excitability.

Finally, while the vestibulospinal tract can influence H-reflexes in the lower extremity [66] its role is minimal during quiet seated postures without significant head motion Thus, the vestibular system likely did not contribute meaningfully to our findings.

### Limitations and future considerations

A key interpretation of our results is that experiencing visual and emotional perturbations in VR enhances fusimotor activity, preserving NTV-reflex strength despite a concurrent increase in spinal inhibition. However, this conclusion is indirect: we inferred fusimotor outflow from reflex behavior rather than measuring spindle feedback directly. Future investigations should strengthen these findings by using microneurography, to monitor individual spindle afferents while ensuring stable α-MN activity, in similar conditions. The challenge of deducing fusimotor drive from muscle spindle activation is well-recognized [10–13].

We must also acknowledge that this study cannot make concrete inferences regarding task or location-specific reflex adaptations. Previous studies have explored lower-limb reflex adaptations during stance [1]; here we investigate upper limb reflexes in a seated posture, where these muscles contribute less to balance and postural correction. Thus, visual or emotional perturbations may have less direct impact on upper limb reflexes than on lower limb muscles that directly maintain stability. Future work could compare both upper and lower limb reflexes within the same experiment to clarify this context dependency.

Finally, while we observed that H-reflexes remained suppressed and EDA remained elevated immediately after VR, the duration of these effect remains unknown. Future studies should explore how long these neural and sympathetic after-effects persist by reassessing H-reflexes and EDA at multiple intervals following VR exposure to monitor their recovery duration and correlation. Future work should also explore how these persistent effects translate to real-world task performance in clinical, occupational, or high-demand environments.

### Conclusion

By elucidating the neural mechanisms that govern motor responses during situations of heightened sympathetic arousal and sensory perturbation such as virtual rollercoaster riding, this research aims to deepen our understanding of human sensorimotor integration and inform the development of interventions aimed at enhancing motor performance and reducing the risk of injury in dynamic environments. Insights gained from this study may have broader implications for fields such as rehabilitation and therapeutics or robotics and prosthetics, where precise control of upper limb movements is essential for optimal performance and safety. It also further highlights the potential for VR as a useful tool for occupational or psychosocial therapies, for example with older adults or in clinical populations such as individuals with anxiety disorders.

The results of this study have important implications for understanding how VR environments affect sensorimotor control and arousal. The sustained suppression of H-reflexes and EDA following VR immersion suggest that prolonged VR exposure may have lingering effects on reflex pathways and sympathetic arousal, which could influence motor performance and proprioception in real-world tasks. This has potential applications in both clinical rehabilitation and virtual training environments, where prolonged VR use may impact motor learning or recovery.

### Supporting information

**S1 File. Spreadsheet containing all the data used to generate the figures and statistical analysis.**
(XLSX)

## Author contributions

**Conceptualization:** Daniel D Hodgson, Brian H Dalton, Tyler Cluff, Ryan M Peters.

**Data curation:** Daniel D Hodgson, Taha Butt.

**Formal analysis:** Daniel D Hodgson, Ryan M Peters.

**Funding acquisition:** Ryan M Peters.

**Methodology:** Ryan M Peters.

**Project administration:** Ryan M Peters.

**Software:** Ryan M Peters.

**Supervision:** Ryan M Peters.

**Visualization:** Daniel D Hodgson, Ryan M Peters.

**Writing – original draft:** Daniel D Hodgson, Tyler Cluff, Ryan M Peters.

**Writing – review & editing:** Daniel D Hodgson, Taha Butt, Brian H Dalton, Tyler Cluff, Ryan M Peters.

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
