## [Decision Letter · Decision Letter 0]

18 Mar 2025

PONE-D-25-07247Upper Limb Muscle Reflexes in Real and Virtual Environments: Insights into Sensorimotor AdaptationsPLOS ONE

Dear Dr. Peters,

Thank you for submitting your manuscript to PLOS ONE. After careful consideration, we feel that it has merit but does not fully meet PLOS ONE’s publication criteria as it currently stands. Therefore, we invite you to submit a revised version of the manuscript that addresses the points raised during the review process.

We look forward to receiving your revised manuscript.

Kind regards,

Hasan Sozen

Academic Editor

PLOS ONE

Journal Requirements:

2**.** Thank you for stating the following financial disclosure:

“This research was funded by Natural Sciences and Engineering Research Council Discovery (RGPIN-2020-06068) and Idea-to-Innovation grants (I2IPJ 548811-2020), as well as a New Frontiers in Research Fund grant (NFRFR-2021-00199) awarded to RMP. DDH was additionally supported through a Natural Sciences and Engineering Research Council Doctoral Postgraduate Scholarship.”

3. In the online submission form, you indicated that “All data will be made available upon request to the corresponding author.”

Reviewers' comments:

Reviewer's Responses to Questions

**Comments to the Author**

1. Is the manuscript technically sound, and do the data support the conclusions?

Reviewer #1: Partly

Reviewer #2: Yes

Reviewer #3: Yes

Reviewer #4: Partly

2. Has the statistical analysis been performed appropriately and rigorously? 

Reviewer #1: No

Reviewer #2: Yes

Reviewer #3: Yes

Reviewer #4: No

3. Have the authors made all data underlying the findings in their manuscript fully available?

Reviewer #1: Yes

Reviewer #2: Yes

Reviewer #3: Yes

Reviewer #4: No

4. Is the manuscript presented in an intelligible fashion and written in standard English?

Reviewer #1: Yes

Reviewer #2: Yes

Reviewer #3: Yes

Reviewer #4: Yes

5. Review Comments to the Author

Reviewer #1: 1. I felt there is too much background information which makes the read very complicated.

2. What tests were done to make sure that the subjects were free from previous musculoskeletal injuries?

3. Age of subjects were approximately 19 years. Why did the researchers not consider other age groups? There should be subjects from different age groups they should consider in their experiments to see differences related to age.

4. What control conditions were used to isolate reflex responses.

Reviewer #2: I would like to express thanks to the authors for presenting rigorous research on sensorimotor adaptations during stressful VR environments.

The authors have presented high-quality work studying the role of visual and emotional on muscle stretch reflex activity in the upper limb. Participants were exposed to a rollercoaster VR environment and measuring the EMG activity in the FCR during isometric contraction. Spinal excitability and fusimotor adaptations were evoked via H-reflex and noisy tendon vibration (NTV) respectively. These measures were used to infer VR-induced central and peripheral sensorimotor adaptations. Consistent with previous studies, stressful environments reduced H-reflex activity in the FCR while maintaining fusimotor strength throughout each condition. However, unlike soleus adaptations, these effects persisted even during the post-VR condition, indicating limb or task specificity.

The results of the paper offers new insights on sensorimotor adaptation in the upper limb during an immersive and visually evoked stressful simulation and builds upon previous research (Hodson et al., 2023). The authors have done exceptionally well in describing their methodology and reasoning.

The claims hinge on the assertion that H-reflex in conjunction with NTV can reliably evoke EMG responses to discern spinal motoneuron excitability and fusimotor sensitivity. One must make several measures to reduce confounding variables such as afferent activity, variable descending and ascending neural activity, and fatigue (both mental and physical). While several controls were included, the difficulty in reliably accessing fusimotor sensitivity transcutaneously remains the biggest weakness. The authors recognise this point and include an in-depth discussion.

Please see the attached pdf for in-line comments.

Minor Issues:

1. I believe line 79 is missing references to research focussing on lower limbs during static balance.

2. While the second paragraph of the introduction points towards the importance of neural adaptation in the upper limbs, further emphasis on why altering visual, sympathetic, and emotional states in the context of sensorimotor adaptation would be appreciated.

3. Line 86 is missing a reference after “…stability and coordinating muscle activity”

4. It would be worthwhile to define ‘SLR’ before being used in line 133. Perhaps in line 131?

5. The first sentence of your discussion section seems slightly disconnected to the paper. Could this be reworded to fully encapsulate the aims?

6. The last two sentences of the paragraph, lines 623 – 627, could be rephrased to have a clearer connection to the observation in the biceps brachii.

7. The logic detailed in lines 702 – 704 could be elaborated upon for clarity.

Comments:

8. Was there a reason why Mmax was stimulated randomly and not near the beginning and end of the trial?

9. Were there any differences in electrophysiological measures in people that expressed stronger motion sickness experience than others?

10. Another consideration for why electrical stimulation events had higher EDA is the stimulation process evoked more sympathetic activity.

11. In line 585-586, “… may reflect a protective mechanism in response to the perceived threat…” is this in relation to the motion sickness experience or the visual experience of being on the virtual rollercoaster itself?

12. Another group to consider the sensorimotor changes observed in your paper, regarding potential stabilisation mechanisms around the upper limb, are athletes such as gymnasts or divers.

13. Another future direction may be to understand underlying spinal reflex and fusimotor adaptations during episodes of motion sickness.

Reviewer #3: The manuscript by Hodgson et al. entitled "Upper Limb Muscle Reflexes in Real and Virtual Environments: Insights into Sensorimotor Adaptations" assesses how virtual reality (VR) influences flexor carpi radialis (FCR) muscle activity. The basic premise, the presented study design, and the applied methodology seem solid. The results are presented in a clear manner. And, the manuscript is generally well written. These positive aspects warrant an overall positive assessment. However, there are a few concerns (outlined below) concerning the lack of envisaged translatability, the statistical analysis, the reproducibility, and the writing style. These concerns should be addressed by the authors. A major revision seems necessary.

General aspects:

1) Writing style

Great parts of the discussion are devoted to summarizing the existing literature. There is an overall lack of interpretation of the results. The discussion should be rewritten with a focus on the authors' own assessments. Since this is not a literature review the discussion could be shortened quite a bit. Furthermore, it should not just repeat the results. The same goes for the limitations' section. Also, the introduction could be shortened and more clear (e.g. lines 170-175 seem not needed; the switch to VR going from line 144 to 145 comes out of the blue).

2) Translatability of findings

Only in the conclusion (e.g. lines 806-806 and 814-815), one finds a potential translatibility of the findings. This should addressed more. The reader needs to understand what potential impact the findings might have and why the study is relevant.

3) Statistical analysis

- A power calculation is missing. This needs to be justified or at least addressed.

- If one uses general linear mixed effects models (lines 404-405) it needs to be clear why this is done. What were the random effects that necessitated the use of such models? This information is completely missing.

- If the motion sickness assessment questionnaire (MSAQ) uses ordinal data (lines 388-393) why is it analysed as if the items were metric (see "Analyzing ordinal data with metric models: What could possibly go wrong?" by Liddell & Kruschke 2018)?

- Considering data points exceeding +/- 2 SDs as outliers and removing them (lines 441-442) needs to be well justified. It is unclear whether the authors checked how such a removal affected their results.

4) Reproducibility

It is unclear what noisy tendon vibrations (NTV) were used. Just giving a frequency range of 0-100Hz (e.g. lines 279 and 352) and no further details makes this intervention completely irreproducible.

In detail:

94-95

"...may arise from..." seems very speculative. Maybe the authors can be more clear here.

121

"...suggesting potential alterations in pre-synaptic inhibition." Where?

154

"Similarly, VR environments..." Similar to what?

178 and 544

There seems to be no secondary objective. So, it does not seem necessary to speak of a "primary" one.

197-204

No prior power analysis is mentioned. If none was done this should be mentioned and justified. Were all subjects right-handed? Was handedness included in the analysis? If not why not?

221

How many seconds passed until the stimulation started (i.e. build-up time before the first drop)?

226-248 and 479-494

It is unclear why the authors assessed the background load level / background muscle activity as there was no hypothesis about this? Was it an exploratory analysis? This should be explained.

230-231

It is commendable that the authors considered that H-reflexes will decrease over time. However, one cannot find how this expectation was included in the statistical analysis. This needs to be clarified.

249

Only in this instance, the authors incorrectly refer to "intensity" instead of current (used everywhere else in the manuscript). This inconsistency should be amended.

327-335

This part does not belong into the methods' section.

362-363

Were impedances measured? Was an impedance cutoff used?

380

"objective measurement" is quite a stretch as it is highly individual. Such a comment is also not necessary in the methods' section.

416

Why was RMS chosen as a quantification metric? Justification and references needed.

417

Why is background EMG mentioned as a dependent variable? There is no intervention on it?

423

What is meant by "using vibration magnitude"?

425-427

Were comparisons to baseline performed?

436-437

What are "strong FCR H-reflexes". Please give mean +/- SD.

443

Were there 17 participants or 18 in total?

448-460

Were H reflex latencies assessed? If not, then this should be justified.

581-582

"...leading to an increase in descending spinal inhibitory commands." References needed for this claim.

584-587

This is a very interesting. Is this an original idea by the authors? If so, this should be stated. Otherwise, references are needed.

607-644

This section could be considerably shortened (especially 608-613).

645-663

This section could be considerably shortened.

696-756

This could be restricted to what is pertinent to the authors' findings.

763-765

This should be in the discussion and not in the limitations.

Reviewer #4: Comments to Author

General comments

The study investigated the modulation of spinal reflex responses using two different methods, H-reflex and NTV, and further examines the neural mechanisms underlying spinal reflex responsiveness by incorporating the indicators of arousal and autonomy function. The research topic is interesting and important for understanding the potential applications of VR technology. While the descriptions, especially in measurement procedures, are detailed, the more critical aspects—such as the analysis steps and statistical methods—are insufficiently detailed, making it hard to determine the validity of the results.

Major comments

1. As the authors know, the peak-to-peak amplitude of Mmax/Hmax varies between individuals. It is crucial to compare H-reflex amplitude while keeping M-wave amplitude constant, which appears to be the rationale for this comparison. However, the amplitude of H-reflex, and possibly the M-reflex, remains in its original unit. It should be normalized using Mmax (%Mmax), and the statistical analysis should be performed again.

2. In the statistical analysis, the LSD method for post-hoc comparisons may not be sufficient. Although the authors are concerned about the small sample size, post-hoc tests, such as Bonferroni correction, are commonly applied even in studies with a similar sample size (n=18). If the statistical approach is considered insufficient, the sample size should be increased based on prior power analysis. Furthermore, the construction of the GLMM model has not been presented and should be explicitly detailed.

3. There is no clear description of the duration for which BGA and EDA were quantified. Since the neural state just before stimulation directly affects responsiveness, the quantification interval should be restricted to this period.

4. The description for experimental procedures are detailed, but they are overly verbose, making it better to simplify them (e.g., the method for the H-reflex). To improve readability, it would be better to reconsider the order of information presentation. In the Introduction section, some parts lack continuity and do not effectively organize key information, such as stretch reflex latency or the explanation of NTV. Furthermore, it is unclear why the study focused on upper limb movement and how the study addresses unresolved issues from previous research. The purpose of the measurements should be clearly stated in the Methods section; however, this information is instead introduced later in the Results section, making the document difficult to follow.

Specific comments

Introduction:

L90, ‘β-MN’: The term β-MN cannot be found in the following text. There is no need to include abbreviations.

L99, ‘noisy tendon vibration (NTV)’: It would be better to include a more detailed explanation, such as the methodology, as an introduction to enhance the reader’s understanding.

L115, ‘as visual feedback.’: References should be placed at the end of the sentence.

L125: The citation for the stretch reflex should include information on its latency and provide an explanation of either SLR, MLR, or LLR.

L127: The section divisions may be incorrect.

L156: The introduction of a study on posture seems abrupt. Given that lack of a clear connection within the context, a more detailed explanation is needed to clarify how this study on standing posture relates to the research objective.

L176: The research topic abruptly shifts to upper limb movement, causing a disruption in continuity. When explaining the objective, it is essential to clarify which previously mentioned issues the study aims to address and how it intends to resolve them.

Methods

L198, ‘using a convenience sampling method’: Please explain in more details.

L216, Figure1: A schematic diagram that illustrates the order and frequency of experimental conditions would enhance clarity. C. It would be better to adjust the scale of the EDA. Additionally, clarification should be provided regarding the meaning of the motor signal command.

L 221: The visibility of the course is limited, and it is unclear what the specific scene the asterisk represents.

L222: It is unclear what Figure C represents. A figure that clearly illustrates the NTV response should be provided.

L 262-263, ‘by the smallest amount of current’: It is inconsistent with the method for identifying H-wave and M-wave described in the following text; therefore, it would be better to delete it.

L277-280: It is unclear how many times NTV was elicited in a single session.

L313-314: Whether the timing of stimulation was consistent across all trials should be stated.

L328-336: It would be better to summarize the explanation in the Introduction and omit it here for readability.

L366: Since it is difficult to accurately infer muscle fiber orientation from the surface, it may be better to modify the expression to a phrase such as ‘relative to the long axis’.

L413 – 421: The explanation of the NTV evaluation method is concise and cites previous literature but lacks sufficient detail. Given the uniqueness of the analysis procedure, a more detailed description should be provided in the paper.

L417: The quantification interval for Background RMS-EEG is not clear defined and should be explicitly stated.

L425: The method for quantifying EDA is unclear. It was stated that measurements were taken throughout the experiment, but was the average calculated across all intervals?

Results

L449: It is Figure 2B, not Figure 4.

L451, ‘22.3%’: The meaning should be explicitly stated. The same applies to all result descriptions that follow.

L458 and 461: The mean and standard error for M-max should be provided for each of the four conditions. Any results not depicted in the graph should be reported with corresponding quantitative values.

L465: Including responses from all four conditions in the example would be preferable. Consider showing the mean for each of the four conditions for a representative participant.

L470: I could not find the 0.001 level in this figure.

L473: It is Figure 3, not Figure 5.

L475: What is the unit of the NTV response?

L482: Results should for H-reflex and NTV should be separated. Additionally, as previously mentioned, the quantification interval is unclear. Given its strong relevance to the response, the quantification should be restricted to the period just before stimulation.

L483: It is Figure 4, not Figure 3.

L491: Presenting BGA and Fatigue level at the same level lacks consistency. Additionally, isn't this redundant with Table 1?

L498: Please specify the quantification interval.

L504: It is Figure 5, not Figure 2.

L504-505 and L507, ‘EDAelevated in all conditions’, ‘… falling just short of significant…’: As the change is not statistically significant, I am afraid that it is misleading.

L522. It is Figure 5, not Figure 2.

L509: Considering the main text, it would be more appropriate to reverse the order of Figures A and B.

Discussion

L569-570: ‘visual perturbation’: The results presented in this section pertain to comparisons of visual conditions without motion; thus, this expression may not be entirely appropriate.

L569-571: Is the influence of arousal not discussed in this section?

L573-576:Clearly delineate which findings are applicable to upper limb muscles and identify the unresolved questions in previous research.

L579: Although many references are cited, it is essential to explicitly state whether each study relates to upper or lower limb tasks.

L580, ‘increased sensory input’: The intended meaning is unclear. Please explicitly specify what other factors, besides 'Visual,' are being referred to, ensuring clarity.

L593 and L594, ’but considering that EDA only marginally…’, ‘elevated symphathetic arousal. ’: No statistically significant difference was observed compared to the baseline. The discussion should be grounded in the statistical results.

L595, ‘visual perturbation’: Does this refer to transient effects associated with returning from VR? Fundamentally, does this not represent a stable visual condition without movement?"

L607: This section discusses the factors contributing to the increase in BB activity. However, the critical issue is whether this outcome influences the primary variable, FCR activity. The discussion should center on aspects that are directly relevant to the study's objective

L656-657: The statistical analysis reveals no significant difference, which undermines the strength of the argument.

L799: Although the significance of the study and future directions are addressed, the most critical aspect—what this study has elucidated and to what extent—remains unaddressed. A clear conclusion summarizing the main findings of the study should be included.

6. PLOS authors have the option to publish the peer review history of their article (what does this mean? ). If published, this will include your full peer review and any attached files.

**Do you want your identity to be public for this peer review?** For information about this choice, including consent withdrawal, please see our Privacy Policy .

Reviewer #1: No

Reviewer #2: No

Reviewer #3: **Yes: ** Alexandros Guekos

Reviewer #4: No

---

## [Author Response · Author response to Decision Letter 1]

18 Jul 2025

Reviewer #1:

1. I felt there is too much background information which makes the read very complicated.

Thanks for this feedback, we have attempted to streamline the background information discussed in the introduction of the updated version.

2. What tests were done to make sure that the subjects were free from previous musculoskeletal injuries?

This was done only via self-report for the present study. All subjects were young, healthy, university-aged students. None of the research team members are clinicians and we did not have access to each participant’s medical records.

3. Age of subjects were approximately 19 years. Why did the researchers not consider other age groups? There should be subjects from different age groups they should consider in their experiments to see differences related to age.

Certainly – in fact this is one of our planned next steps for this work. We have a hypothesis that the effects observed in the present study will be exacerbated in an older adult cohort, but this remains to be tested.

4. What control conditions were used to isolate reflex responses.

Reflex responses were isolated in our protocol by virtue of each reflex (tendon vibrations and H-reflexes) being tested separately, and not concomitantly. We hope that the timeline for stimulus presentations is clarified further now by the addition of Figure 1B, which is a schematic depiction of the experimental protocol. Also, as a control or “baseline” condition, we assessed reflexes both before and after VR exposure, and the order of presentation was randomized across participants.

Reviewer #2: I would like to express thanks to the authors for presenting rigorous research on sensorimotor adaptations during stressful VR environments.

The authors have presented high-quality work studying the role of visual and emotional on muscle stretch reflex activity in the upper limb. Participants were exposed to a rollercoaster VR environment and measuring the EMG activity in the FCR during isometric contraction. Spinal excitability and fusimotor adaptations were evoked via H-reflex and noisy tendon vibration (NTV) respectively. These measures were used to infer VR-induced central and peripheral sensorimotor adaptations. Consistent with previous studies, stressful environments reduced H-reflex activity in the FCR while maintaining fusimotor strength throughout each condition. However, unlike soleus adaptations, these effects persisted even during the post-VR condition, indicating limb or task specificity.

The results of the paper offers new insights on sensorimotor adaptation in the upper limb during an immersive and visually evoked stressful simulation and builds upon previous research (Hodson et al., 2023). The authors have done exceptionally well in describing their methodology and reasoning. Thanks!

The claims hinge on the assertion that H-reflex in conjunction with NTV can reliably evoke EMG responses to discern spinal motoneuron excitability and fusimotor sensitivity. One must make several measures to reduce confounding variables such as afferent activity, variable descending and ascending neural activity, and fatigue (both mental and physical). While several controls were included, the difficulty in reliably accessing fusimotor sensitivity transcutaneously remains the biggest weakness. The authors recognise this point and include an in-depth discussion.

Please see the attached pdf for in-line comments.

Minor Issues:

1. I believe line 79 is missing references to research focussing on lower limbs during static balance.

Good catch – we have added a reference here.

2. While the second paragraph of the introduction points towards the importance of neural adaptation in the upper limbs, further emphasis on why altering visual, sympathetic, and emotional states in the context of sensorimotor adaptation would be appreciated.

Agreed – as part of streamlining of the intro based on Reviewer 1’s comment, we have also refocussed the intro with added emphasis on the upper limb.

3. Line 86 is missing a reference after “…stability and coordinating muscle activity”

We have added a reference for this statement.

4. It would be worthwhile to define ‘SLR’ before being used in line 133. Perhaps in line 131?

Good call, we have added a definition of SLR there.

5. The first sentence of your discussion section seems slightly disconnected to the paper. Could this be reworded to fully encapsulate the aims?

Agreed – we have reworked that sentence to better reflect the aims of our study.

6. The last two sentences of the paragraph, lines 623 – 627, could be rephrased to have a clearer connection to the observation in the biceps brachii.

We have taken your advice and reworded these sentences as well – fair point.

7. The logic detailed in lines 702 – 704 could be elaborated upon for clarity.

This has been further clarified in the updated manuscript.

Comments:

8. Was there a reason why Mmax was stimulated randomly and not near the beginning and end of the trial?

From our experience, the Mmax stimulation level can be a bit more startling/uncomfortable for the participants. So our logic here was to space them out and to apply them at random times so the participants would not make any anticipatory adjustments to their posture, muscle activation, etc.

9. Were there any differences in electrophysiological measures in people that expressed stronger motion sickness experience than others?

We have explored this further; however, we do not pick up on any obvious trends in the data that would suggest other differences explained stronger motion sickness scores.

10. Another consideration for why electrical stimulation events had higher EDA is the stimulation process evoked more sympathetic activity.

Agreed – we have added further interpretation of this finding in the revised manuscript.

11. In line 585-586, “… may reflect a protective mechanism in response to the perceived threat…” is this in relation to the motion sickness experience or the visual experience of being on the virtual rollercoaster itself?

Good point – we have added further description in the manuscript for what we believe is the source of the underlying threat.

12. Another group to consider the sensorimotor changes observed in your paper, regarding potential stabilisation mechanisms around the upper limb, are athletes such as gymnasts or divers.

Agreed – we believe this is an excellent future direction for this work and are planning to investigate further!

13. Another future direction may be to understand underlying spinal reflex and fusimotor adaptations during episodes of motion sickness.

This is an interesting point – our lab is beginning new projects on individuals with motion sickness and this would make for a very neat follow up study. Thanks for this suggestion!

Reviewer #3: The manuscript by Hodgson et al. entitled "Upper Limb Muscle Reflexes in Real and Virtual Environments: Insights into Sensorimotor Adaptations" assesses how virtual reality (VR) influences flexor carpi radialis (FCR) muscle activity. The basic premise, the presented study design, and the applied methodology seem solid. The results are presented in a clear manner. And, the manuscript is generally well written. These positive aspects warrant an overall positive assessment. However, there are a few concerns (outlined below) concerning the lack of envisaged translatability, the statistical analysis, the reproducibility, and the writing style. These concerns should be addressed by the authors. A major revision seems necessary.

General aspects:

1) Writing style

Great parts of the discussion are devoted to summarizing the existing literature. There is an overall lack of interpretation of the results. The discussion should be rewritten with a focus on the authors' own assessments. Since this is not a literature review the discussion could be shortened quite a bit. Furthermore, it should not just repeat the results. The same goes for the limitations' section. Also, the introduction could be shortened and more clear (e.g. lines 170-175 seem not needed; the switch to VR going from line 144 to 145 comes out of the blue).

Agreed – we have rewritten and streamlined large portions of the discussion to address this point, adding more interpretation and reducing review of existing literature. Given comments from Reviewer’s 1 and 2, we have also streamlined and refocussed the introduction as you have suggested.

2) Translatability of findings

Only in the conclusion (e.g. lines 806-806 and 814-815), one finds a potential translatibility of the findings. This should addressed more. The reader needs to understand what potential impact the findings might have and why the study is relevant.

Thanks, we agree that we have perhaps understated the applicability of our findings and have tried to bring that to the forefront more in the revised manuscript.

3) Statistical analysis

- A power calculation is missing. This needs to be justified or at least addressed.

We did not perform any formal power analysis in this case, rather our sample size was selected based on previous sample sizes used for studies investigating H-reflexes during exposure to VR, which ranged from 15 (Grospretre et al., 2023) to 20 (Hodgson et al., 2023). The results from this paper will be used in future work as a basis for a more formal a priori power calculation. We now explicitly discuss this in the revised manuscript.

- If one uses general linear mixed effects models (lines 404-405) it needs to be clear why this is done. What were the random effects that necessitated the use of such models? This information is completely missing.

The random effects in this case are necessitated by the degree of inter-subject variability in reflex responses we commonly observe, which can be accounted for using a linear mixed effects model. We have attempted to better justify the use of this statistical approach in the revised manuscript.

- If the motion sickness assessment questionnaire (MSAQ) uses ordinal data (lines 388-393) why is it analysed as if the items were metric (see "Analyzing ordinal data with metric models: What could possibly go wrong?" by Liddell & Kruschke 2018)?

Thank you for raising this concern. However, we note that, while each individual item on the MSAQ is indeed ordinal in nature, what gets used in the analysis is actually composite scores that are expressed as a percentage of total points scored from all 16 items. This is why MSAQ scores are treated as metric data in our analysis.

- Considering data points exceeding +/- 2 SDs as outliers and removing them (lines 441-442) needs to be well justified. It is unclear whether the authors checked how such a removal affected their results.

+/- 2 SD was chosen as this is a commonly used cutoff for outlier analysis. We did indeed run our analysis without removing outliers which had no effect on our conclusions/trends. We also note that this removal was only for a single subject with irregularly high values.

4) Reproducibility

It is unclear what noisy tendon vibrations (NTV) were used. Just giving a frequency range of 0-100Hz (e.g. lines 279 and 352) and no further details makes this intervention completely irreproducible.

Good point - we have added further description of the vibration stimulus used in this study.

In detail:

94-95

"...may arise from..." seems very speculative. Maybe the authors can be more clear here.

Agreed – we have reworded this statement accordingly.

121

"...suggesting potential alterations in pre-synaptic inhibition." Where?

Added more detail here.

154

"Similarly, VR environments..." Similar to what?

Reworded this sentence.

178 and 544

There seems to be no secondary objective. So, it does not seem necessary to speak of a "primary" one.

Agreed – we have rephrased this.

197-204

No prior power analysis is mentioned. If none was done this should be mentioned and justified. Were all subjects right-handed? Was handedness included in the analysis? If not why not?

Regarding power analysis, please see our reply to review 2 above – we have now explicitly addressed this in the manuscript as well. The choice in choosing the right arm for testing was based solely on the limitation that our equipment was setup to only stimulate/support the right arm. There is no literature on handedness effect in VR studies like this, however, this is certainly something we could look at in the future.

221

How many seconds passed until the stimulation started (i.e. build-up time before the first drop)?

As described in the methods, the build-up time before cresting the first hill was approximately 60s. We’d also point out that the stimulation begins at the position on the rollercoaster track denoted by a large asterisk in Figure 1C.

226-248 and 479-494

It is unclear why the authors assessed the background load level / background muscle activity as there was no hypothesis about this? Was it an exploratory analysis? This should be explained.

Good point – essentially, background muscle activity is known to affect reflex response amplitudes, so we wanted to ensure that this was not changing across testing conditions for the target muscle. Hence, showing no change in muscle activity level demonstrates that this was held constant and could not have been the cause of any changes observed in the reflex responses.

230-231

It is commendable that the authors considered that H-reflexes will decrease over time. However, one cannot find how this expectation was included in the statistical analysis. This needs to be clarified.

This was concluded based on the main-effect and subsequent post-hoc comparisons in the statistical analysis where we show that H-reflex amplitude decreased significantly in the VR-C condition and remained depressed in the VR-POST condition. However, your point about how this expectation was incorporated is well-taken – we do indeed have directional predictions for EDA and H-reflexes based on the previous literature, therefore, in the updated manuscript we have switched to one-tailed (directional) post-hoc comparisons for these variables. Thank you for pointing this out!

249

Only in this instance, the authors incorrectly refer to "intensity" instead of current (used everywhere else in the manuscript). This inconsistency should be amended.

Agreed – we have changed this heading to “Determination of Electrical Stimulation Current”.

327-335

This part does not belong into the methods' section.

Agreed – we have removed this from the methods section.

362-363

Were impedances measured? Was an impedance cutoff used?

We did not measure impedances in the present study. However, the stimulation module we used (Biopac, STMISOLA) is a constant current source, meaning that any variations in impedance are counteracted by changing the voltage such that the commanded current is accurately delivered. Additionally, the skin was prepped the same in all cases and electrodes were all newly purchased and fresh.

380

"objective measurement" is quite a stretch as it is highly individual. Such a comment is also not necessary in the methods' section.

Agreed – we have removed this editorialization from the methods.

416

Why was RMS chosen as a quantification metric? Justification and references needed.

Agreed – we have added this information. RMS is a commonly used measure of EMG magnitude and is consistent with reporting in our previous paper (Hodgson et al, 2023).

417

Why is background EMG mentioned as a dependent variable? There is no intervention on it?

As mentioned above, it is critical to measure and control background muscle activity in the target muscles as this is known to affect reflex response amplitudes. We needed to confirm that this variable did not change significantly across our testing conditions.

423

What is meant by "using vibration magnitude"?

As with our previous study (Hodgson et al 2023), we used a three-axis accelerometer to capture any off-normal-axis vibrator motion, and for the analysis we computed vibration magnitude by summing the squared X, Y, and Z axis accelerations, and taking the square root. This expresses the vibration acceleration as

---

## [Decision Letter · Decision Letter 1]

27 Jul 2025

Upper Limb Muscle Reflexes in Real and Virtual Environments: Insights into Sensorimotor Adaptations

PONE-D-25-07247R1

Dear Dr. Peters,

We’re pleased to inform you that your manuscript has been judged scientifically suitable for publication and will be formally accepted for publication once it meets all outstanding technical requirements.

Kind regards,

Hasan Sozen

Academic Editor

PLOS ONE

Reviewers' comments:

Reviewer's Responses to Questions

**Comments to the Author**

1. If the authors have adequately addressed your comments raised in a previous round of review and you feel that this manuscript is now acceptable for publication, you may indicate that here to bypass the “Comments to the Author” section, enter your conflict of interest statement in the “Confidential to Editor” section, and submit your "Accept" recommendation.

Reviewer #1: All comments have been addressed

Reviewer #2: All comments have been addressed

Reviewer #3: All comments have been addressed

2. Is the manuscript technically sound, and do the data support the conclusions?

Reviewer #1: Yes

Reviewer #2: Yes

Reviewer #3: Yes

3. Has the statistical analysis been performed appropriately and rigorously? 

Reviewer #1: Yes

Reviewer #2: Yes

Reviewer #3: Yes

4. Have the authors made all data underlying the findings in their manuscript fully available?

Reviewer #1: Yes

Reviewer #2: Yes

Reviewer #3: Yes

5. Is the manuscript presented in an intelligible fashion and written in standard English?

Reviewer #1: Yes

Reviewer #2: Yes

Reviewer #3: Yes

6. Review Comments to the Author

Reviewer #1: The authors have addressed the question I had asked and corrected them. Except the broader age group needs to be considered, which they plan to do in their next research.

Reviewer #2: Thank you to the authors for their input and addressing each comment. I am satisfied with this revision.

Reviewer #3: The authors have substantially rewritten their manuscript. They have addressed all issues raised in a satisfactory manner. Therefore, the present revised version may be accepted without reservations.

7. PLOS authors have the option to publish the peer review history of their article (what does this mean? ). If published, this will include your full peer review and any attached files.

**Do you want your identity to be public for this peer review?** For information about this choice, including consent withdrawal, please see our Privacy Policy .

Reviewer #1: No

Reviewer #2: No

Reviewer #3: **Yes: ** Alexandros Guekos

---

## [Editor Report · Acceptance letter]

PONE-D-25-07247R1

PLOS ONE

Dear Dr. Peters,

I'm pleased to inform you that your manuscript has been deemed suitable for publication in PLOS ONE. Congratulations! Your manuscript is now being handed over to our production team.

Kind regards,

on behalf of

Assoc. Prof. Hasan Sozen

Academic Editor

PLOS ONE